# Vitiligo, from Pathogenesis to Therapeutic Advances: State of the Art

**DOI:** 10.3390/ijms24054910

**Published:** 2023-03-03

**Authors:** Federico Diotallevi, Helena Gioacchini, Edoardo De Simoni, Andrea Marani, Matteo Candelora, Matteo Paolinelli, Elisa Molinelli, Annamaria Offidani, Oriana Simonetti

**Affiliations:** Dermatological Clinic, Department of Clinical and Molecular Sciences, Polytechnic Marche University, 60121 Ancona, Italy

**Keywords:** vitiligo, pathogenesis, treatments, afamelanotide, prostaglandin, Janus kinase inhibitors

## Abstract

Vitiligo is an acquired hypopigmentation of the skin due to a progressive selective loss of melanocytes; it has a prevalence of 1–2% and appears as rounded, well-demarcated white macules. The etiopathology of the disease has not been well defined, but multiple factors contribute to melanocyte loss: metabolic abnormalities, oxidative stress, inflammation, and autoimmunity. Therefore, a convergence theory was proposed that combines all existing theories into a comprehensive one in which several mechanisms contribute to the reduction of melanocyte viability. In addition, increasingly in-depth knowledge about the disease’s pathogenetic processes has enabled the development of increasingly targeted therapeutic strategies with high efficacy and fewer side effects. The aim of this paper is, by conducting a narrative review of the literature, to analyze the pathogenesis of vitiligo and the most recent treatments available for this condition.

## 1. Introduction

Vitiligo is a relatively common acquired pigmentation disorder affecting about 1–2% of the world’s population, characterized by the development of well-defined depigmented macules on the skin that reflect the loss of epidermal melanocytes [1]. Lesions may occur in a localized or generalized distribution and may aggregate into large, depigmented areas. Given the contrast between white areas and normal skin, the disease deeply impacts the quality of life of children and adults by causing stigmatization, social isolation, and low self-esteem in affected individuals [2,3,4].

The treatment of vitiligo is not simple and is based on the spread and localization of the lesion and whether the disease is more or less stabilized. In addition to the traditional treatments consisting of phototherapy and topical or systemic steroids, years of research into pathogenesis have led to the development of more targeted therapies with high efficacy and reduced side effects [5]. The aim of this paper is, by conducting a narrative review of the literature, to analyze the pathogenesis of vitiligo and the most recent treatments available for this condition.

## 2. Materials and Methods

To conduct this narrative review, four key points typical of biomedical narrative reviews were followed: 1—identify keywords, 2—conduct research, 3—review abstract and articles, and 4—document results [6].

### 2.1. Identify Keywords

To identify keywords, a brainstorming approach involving the entire research group was used. The research team consisted of three dermatologists with expertise in vitiligo pathophysiology and three dermatologists with expertise in vitiligo therapy. Two other dermatologists with good knowledge of the disease also had specific expertise in literature review methodology.

In the first meeting, the research team selected the topic, identified the scope, constructed the title, and chose the keywords as follows: “Vitiligo”, “Vitiligo and Pathogenesis”, “Vitiligo and Treatments”, “Vitiligo and Afamelanotide”, “Vitiligo and Prostaglandins”, “Vitiligo and Janus kinase Inhibitors”, and “Vitiligo and JAK inhibitors”.

### 2.2. Conduct Research

Articles were searched in the following databases: National Library of Medicine PubMed database and Scopus. The reference lists of the selected studies, especially in the cases of the reviews, were scanned for additional papers.

Inclusion criteria: studies reporting on vitiligo, published in English language, and published between 1990 and 2022, with an abstract available. No restriction on the design of the study was considered, and randomized controlled trials, case–control studies, cross-sectional studies, case reports and series, and review articles were included.

### 2.3. Review Abstract and Article

The selection of the relevant data published in the literature took place in three steps. In the first step, three researchers (F.D., M.P. and O.S.) independently selected the articles based on the title. Any disagreement was solved by consulting a senior investigator (A.O.). The second step consisted of evaluating the abstracts. At least four members of the research team (H.G., E.D.S., A.M., and M.C.) independently assessed each abstract. The research team resolved all discrepancies through consensus. Linguistic revision was performed by E.M.

### 2.4. Document Results

All sources with similar data/level of evidence were analyzed, collected, and grouped. The main text was structured into subsections. New evidence-based points were summarized, and the major points for future research and practice were defined.

## 3. Results

Two macro topics emerged from the literature review: “Pathogenesis of Vitiligo” and “Treatments of Vitiligo: past, present, and future”.

### 3.1. Pathogenesis of Vitiligo

Vitiligo is a multifactorial disorder characterized by the destruction of functional epidermal melanocytes [7]. The precise etiology and pathophysiology are complex, and there is still much debate about the various theories on the loss of melanocyte function. Multiple pathophysiological mechanisms, including genetics, autoimmunity, oxidative stress, and neurological system disfunction, have been proposed [8]. Thus, different vitiligo phenotypes cannot be explained by only one of these mechanisms, and disparate mechanisms might contribute to the same clinical result [9].

Therefore, the convergence theory was proposed that combines all existing theories into a comprehensive one in which several mechanisms contribute to the reduction of melanocyte viability. [9]. Although each of these pathogenetic hypotheses are still under discussion, there is now an agreement on the autoimmune and oxidative stress theories as leading processes in vitiligo pathogenesis [10].

The pathogenetic mechanisms of vitiligo are illustrated in Figure 1.

#### 3.1.1. Genetics

Strong evidence from multiple studies supports the multifactorial, polygenic inheritance of vitiligo [11,12,13]. Genetic and environmental factors are estimated to have a risk of about 80% and 20%, respectively. [14]. Several epidemiological studies on vitiligo revealed that this disease can cluster within families [12,13]. Thus, overall, about 20% of patients report at least an affected first-degree relative, and the concordance rate in monozygotic twins is around 23% [11].

Common genetic variants (risk allele frequency > 0.01) represent about 71% of the total vitiligo heritability and about 53% of the total vitiligo risk, whereas the remaining 29% of heritability and 23% of the entire risk can be attributed to rare variants [15].

A DNA sequence analysis is the best resource to understand the complex polygenic inheritance of vitiligo. Today, Genome-Wide Association Studies (GWAS), conducted so far on European and Asian populations, have identified at least 54 vitiligo susceptibility loci [15,16,17,18,19]. The majority of them are involved in regulation of the immune system; the recognition and apoptosis of melanocytes; and several are shared with other autoimmune diseases such as thyroid disease, type 1 diabetes, and rheumatoid arthritis [20]. An exception is represented by the TYR gene, which encodes for tyrosinase, a key enzyme in melanin biosynthesis that has been found in a GWAS performed on European White people with nonsegmental vitiligo [19,21].

A genome-wide linkage analysis detected seven alleged vitiligo susceptibility loci, of which five have been associated with a causal gene: FOXD3 (Forkhead Box D3), NLRP1 (NLR family pyrin domain containing 1), PDGFRA (Platelet-Derived Growth Factor Receptor Alpha), HLA (human leukocyte antigen), and XBP1 (X-box binding protein 1) [22].

FOXD3 mutations, resulting in transcriptional upregulation, seem to interfere with melanoblast differentiation [23]. NLRP1 is a key regulator of the innate immune system, especially in the skin where, in response to specific triggers, it activates the inflammasome and processes pro-Interleukin 1 beta (IL1β) into its active form (IL1β), which regulates the polarization of T cells towards Th17, thus perpetuating the inflammatory response [24].

Although the presence of Th17 cells and high levels of IL17 are established in vitiligo, as in other diseases that have an autoimmune component, the role these cells play in the pathogenesis of the disease is still unclear. Several studies have also demonstrated a significant correlation between the serum levels of Th17 and IL17 with the disease duration, extent, and activity [25]. Furthermore, there are studies showing that NB-UVB, through a reduction in IL-17 expression, improves vitiligo lesions [26]. Several hypotheses have been made on how IL17 produced by Th17 cells may contribute to disease development. The first theory involves the chemokine CCL20 produced by IL-17. Indeed, this cytokine, being a homing molecule, would attract CD8+ T cells that have been implicated in the direct killing of melanocytes in vitiligo models [27]. Another way in which IL-17 could be involved is through the stimulation of the endothelial expression of E- and P-selectins and the adhesion molecules ICAM-1 and VCAM-1, resulting in the migration of neutrophils, which would favor the production of several ROS crucial in the destruction of melanocytes [28].

The PDGFRA gene exerts an essential role in the differentiation and survival of melanocytes during embryonic development and in the regulation of pigmentation and is associated with defective melanocyte migration [29].

Vitiligo-associated HLA alleles may not be disease-specific. HLA alleles implicated in vitiligo and found so far are HLA-DRB4*0101, HLA-DQB1*0303, HLA-DRB1*03, HLA-DRB1*04, HLA-DRB1*07, and HLA DRB1A*04-(DQA1*0302)-DQB1*0301 [30].

XBP1 is a transcription factor that may influence the development of vitiligo through its interaction with HLA-DR molecules [31].

Many other genes are implicated in vitiligo pathogenesis such as CTLA-4 (Cytotoxic T-Lymphocyte Antigen 4), ACE (Antigen converting enzyme), CAT (Catalase), PTPN22 (Protein Tyrosine Phosphatase Non-Receptor Type 22), MYG1 (MYG1 Exonuclease), MITF (Melanocyte Inducing Transcription Factor), KIT (KIT Proto-Oncogene, Receptor Tyrosine Kinase), ESR 1 (Estrogen Receptor 1), AIRE (Autoimmune Regulator), COMT (Catechol-O-methyltransferase), NALP1 (Nucleotide-binding oligomerization domain, Leucine rich Repeat and Pyrin domain containing), FAS (Fas Cell Surface Death Receptor), EDN1 (Endothelin 1), COX2 (Cyclooxygenase 2), VIT1 (Vacuolar iron transporter 1), IKZF4 (IKAROS Family Zinc Finger 4), and DDR1 (Discoidin Domain Receptor Tyrosine Kinase 1) [30,32]. Recently, an increasing body of evidence has been observed for the correlation between autoimmune disorders and vascular endothelial growth factor (VEGF) polymorphism [33,34]. Vascular endothelial growth factor (VEGF) is one of primary regulators of angiogenesis studied in different chronic diseases and cancers [35,36,37,38]. In vitiligo, a significant association between the GG genotype and higher age at onset of the disease has been observed (*p* = 0.04) [39].

#### 3.1.2. Autoimmunity

Genetic studies confirm the autoimmune hypothesis as the vitiligo leading pathogenetic mechanism, since around 85% of the vitiligo susceptibility genes encode molecules implicated in innate and adaptive immunity and apoptosis [40].

The autoimmune theory is also supported by the association of vitiligo with other autoimmune disorders, the presence in vitiligo patients of organ-specific antibodies and, indirectly, the immune-modulating nature of vitiligo therapies [10,41].

In addition, the involvement of autoimmunity in the development of vitiligo is confirmed by the observation that, in neoplastic subjects treated with checkpoint inhibitors, the disease can develop. Thus, it has been observed that, in subjects with metastatic melanoma treated with the cytotoxic T-lymphocyte antigen-4 (CTLA-4) pathway inhibitor and programmed cell death-1 protein (PD-1) pathway inhibitor, increasing T cells, the development of vitiligo could also potentially lead to better response and survival rates [42]. In contrast, Miao et al., using a transgenic mouse model of vitiligo with T-cell receptor Pmel-1, showed that the PD-L1 fusion protein reduced the number of melanocyte-responsive T cells, inhibited the activation of Vβ12-expressing T cells, and increased the number of Tregs, reversing depigmentation [43]. Nevertheless, PD-L1 treatment may still necessitate prolonged treatment with NB-UVB therapy, which probably upregulates PD-L1 expression in an NF-κB-dependent way, thus indicating that the use of combined local therapy with PD-1/PD agonist treatment and NB-UVB therapy is a hopeful choice [44].

Several studies have been conducted on vitiligo probands and their relatives, revealing the association of vitiligo with immune-mediated diseases such as autoimmune thyroid disorders, Addison’s disease, pernicious anemia, alopecia areata, rheumatoid arthritis, psoriasis, diabetes mellitus, and systemic lupus erythematosus [12,45,46,47].

One of the most significant studies ever performed, inspecting 2624 vitiligo probands, showed that 19.4% of patients (aged > 20 years) reported a personal history of autoimmune thyroid disease (most commonly hypothyroidism), an eightfold increase over the 2.39% population frequency [45].

Thus, both innate and adaptive immunity take part in vitiligo pathogenesis. Indeed, innate immune cells seem to be early activated by endogenous and exogenous stress signals released from melanocytes and keratinocytes, leading to the subsequent activation of an adaptive immune response, both humoral and cell-mediated, causing the targeted autoimmune destruction of melanocytes [48].

Melanocytes use exosomes, extracellular vesicles that contain miRNAs, melanocyte-specific antigens, heat shock proteins, and other molecules acting as damage-associated molecular patterns (DAMPs), to communicate stress to the innate immune system and, in particular, to dendritic cells that function as antigen-presenting cells [49,50,51]. Inducible heat shock protein 70 (HSP70i), represents one of these DAMPs and, in the mouse model, has demonstrated inducing dendritic cells to present melanocyte-specific antigens to T lymphocytes in lymphoid tissues [52]. The HSP70i importance in vitiligo is also confirmed in another animal models in which a modified HSP70i, Hsp70iQ435A, introduced through the DNA jet injection of a plasmid, was found to produce the re-pigmentation of vitiligo lesions in a long-lasting way, thus representing a potential new treatment for vitiligo [53].

Cluster differentiation (CD)8+ T cells are crucial for the destruction of melanocytes in vitiligo lesions. These cells have been isolated in higher numbers in the blood and in lesional and perilesional vitiligo patients [53,54,55]. Furthermore, their number seems to correlate with disease activity [56]. The antigens targeted by these autoreactive melanocyte-specific cytotoxic lymphocytes derived from proteins of the melanogenic pathway, such as gp100, (Melanoma antigen recognized by T cells 1) MART1, tyrosinase, and tyrosinase-related proteins 1 and 2, thus mediating melanocyte destruction [57].

CD8+ T cells produce interferon-γ (IFN-γ), which is responsible for CD8+ T-cell recruitment in lesional skin through positive feedback thanks to CXC chemokine ligand 9 (CXCL9), CXCL10, and CXCL11, chemokines that mediate T-cell homing to the epidermis. These chemokines are indeed increased in the sera and lesional vitiligo skin and correlate with disease activity and severity [58,59,60].

IFN-γ binds to its cell surface receptor (IFNgR), causing the recruitment of Janus kinase (JAK)-1 and JAK2 that, through the phosphorylation of signal transducer and activator of transcription (STAT), lead to the transcription of IFN-γ-inducible genes. JAK1 expression is higher in the lesional skin of vitiligo patients, and it correlates with a lower percentage of surviving melanocytes. All these findings support the investigation of therapies targeting JAK1 and JAK2 [61,62].

Moreover, in vitiligo patients, there is an impairment of regulatory T cells (Tregs), which suppress the proliferation and activation of autoreactive effectors such as CD8+. The number of Tregs and molecules that support their function, such as homing receptor C-C Motif Chemokine Ligand 22 (CCL22), were found diminished in vitiligo lesions. Conversely, an increased expression of CCL22 can stimulate Treg skin homing, thus impeding the depigmentation process [63,64,65].

Vitiligo presents an estimated risk of 40% of relapse after re-pigmentation within the first year [66]. It has been suggested that relapse is due to persistency in the vitiligo skin of CD8+ tissue-resident memory T cells (Trm), which maintenance and function are promoted by IL-15. Trm cells in vitiligo patients were found to exhibit high levels of CD122, the subunit of the IL-15 receptor, both in blood and lesional skin [67,68]. Furthermore, the anti-CD122 antibody has demonstrated reverse of the disease in mice, so targeting IL-15 signaling through the anti-CD122 antibody could represent a possible successful and durable treatment for vitiligo [69].

As for humoral immunity, patients with vitiligo were found to have elevated serum titers of antibodies directed against melanocytes (e.g., anti-MelanA, anti-Melanin Concentrating Hormone Receptor 1 (MCHR1), anti-tyrosinase, anti-gp100, and anti-tyrosine hydroxylase) that do not correlate with vitiligo activity, and they cannot be considered the main driver of vitiligo pathogenesis [70].

The importance of autoimmunity and genetics in the disease’s development is demonstrated by the presence of several preclinical studies involving IL-15 pathways; studies targeting the production of ROS, DAMPs, and antioxidant pathways or chemokine receptors; and stimulating melanocyte stem cells [71]. These include, firstly, a preclinical study in a porcine vitiligo model that demonstrated the blockade of one of the DAMPs associated with vitiligo; secondly, a preclinical study in which IL-15 was administered to a mouse model of vitiligo; and, thirdly, a preclinical study targeting the chemokine receptor CXCR3 [53,69]. Lastly, preclinical studies targeting Treg in the skin using CCL22 have also shown good efficacy [65]. Unfortunately, many preclinical studies have yet to be followed up in the clinical setting.

#### 3.1.3. Oxidative Stress Hypothesis

Oxidative stress plays a crucial role in initiating vitiligo with melanocyte destruction.

Thus, cutaneous melanocytes, which are also exposed to ultraviolet (UV) radiation and chemical pollutants, are more susceptible to both excessive production and inadequate scavenging of reactive oxygen species (ROS) as a respond to stress. This results in an extensive alteration of the oxidative/antioxidative balance [72]. Notably, there is an increase in prooxidants such as superoxide dismutase, malondialdehyde, and xanthine oxidase and a decrease in antioxidants such as catalase, glutathione reductase, glutathione peroxidase, and superoxide dismutases, as demonstrated in skin and blood [73,74].

Endogenous and exogenous stressor can trigger the excessive formation of ROS. Exogenous stressors include environmental triggers (e.g., UV, cytotoxic chemicals, and trauma); medications (e.g., drugs, hormones, and vaccination); and internal disorders (malignancies, infections, neural disorders, and calcium imbalance). Conversely, excessive ROS production can come from inner stimuli such as melanogenesis, an energy-consuming process that require large amounts of ATP and abnormal mitochondrial energy metabolism [57,75,76,77]. Indeed, the production of a considerable number of proteins during melanogenesis augments the risk of proteins misfolding, causing the activation of unfolded protein response (UPR). XBP1, encoding X-box binding protein 1, is one of the crucial UPR components that sets in motion the production of immune mediators such as IL-6 and IL-8 that halt Tregs and recruit other immune cell populations [78,79].

H_2_O_2_ induces the production of transient receptor potential cation channel subfamily M member 2 (TRPM2), a calcium channel sensitive to oxidative stress resulting in an increased flow of mitochondrial calcium, thus, causing mitochondria-dependent apoptosis of melanocytes [80].

Melanocytes seem to be both the instigators and the victims of oxidative stress. Melanocytes from patients with vitiligo are more susceptible to oxidative stress than those from controls without vitiligo. This concept is corroborated by the fact that vitiligo patients’ melanocytes display a greater difficulty to culture ex vivo than those from healthy controls, and by the lower antioxidant capacity in vitiligo patients, consisting in glutathione peroxidase reduction, when compared to individuals without vitiligo [81,82].

Vitiligo melanocytes present an impairment of nuclear factor erythroid2-related factor (Nrf2), a major pathway in the cellular defense against oxidative stress. Nrf2 transcriptional activity was found to be significantly lower in vitiligo melanocyte cell line than in normal human melanocytes [83]. Known activators of Nrf2 are dimethyl fumarate (DMT) and afamelanotide. DMT activates Nrf2 by increasing nuclear Nrf2 localization, and afamelanotide mimics α-melanocyte-stimulating hormone (MSH) which can increase Nrf2, thus representing promising therapeutic strategies [84].

As demonstrated in a murine model, ROS accumulation also leads to an impairment of the autophagy process, consisting of degradation of damaged organelles and proteins, essential to maintain cellular homeostasis [85].

Redox imbalance is characterized by glutathione depletion, which in turn causes an H2O2-induced decrease in PGE2 synthesis. Hence, prostaglandin analogues can be a favorable alternative treatment for vitiligo patients [86].

Finally, oxidative stress seems to be partially implicated in vitiligo’s Köbner phenomenon. Chronic friction brings inflammatory mediators release, which leads to an increase in ROS and adhesion defects in melanocytes. Moreover, epithelial trauma induces the release of IFNα that, in turn, increment the expression of CXCL10, promoting the migration of circulating lymphocytes to the skin [87,88].

Treatment with narrowband UVB (NB-UVB), one of the commonest vitiligo therapies, can improve oxidant–antioxidant imbalance in vitiligo patients, as shown by the reduction in levels of malondialdehyde, a prooxidant, and by the increase in glutathione peroxidase levels, an antioxidant agent [89].

On the other hand, there is no solid evidence about the use of topical and oral antioxidants in vitiligo patients. This could be explained by limited patient number of studies conducted so far on this topic, by insufficient dosing of antioxidants or by the fact that oxidative stress may not play a central role in vitiligo pathogenesis [90].

Another important mechanism in which oxidative stress would contribute to the onset of vitiligo is the loss of melanocyte dendrites, resulting in the failure of melanin transfer to keratinocytes. Thus, there are studies showing that α-MSH, used in the treatment of vitiligo, protects against H2O2-induced loss of dendrites in melanocytes through activation of the mTORC1 (mammalian target of rapamycin complex 1) pathway, which is inhibited by rapamycin [91].

However, these studies were disproved by subsequent research in which inhibition of the mTOR pathway was useful in inducing vitiligo re-pigmentation. It would seem that, similar to lupus erythematosus, the mTOR pathway is responsible for Treg depletion, and consequently inhibition of this pathway could have benefits in the therapeutic management of patients [92]. Indeed, rapamycin, an inhibitor of PI3Kakt (phosphatidylinositol 3-kinase)-mTORC1 signaling, appears to increase Treg lymphocytes in h3TA2 mice, arresting the depigmentation process in vitiligo [93]. A phase 2 clinical study (ClinicalTrials.gov Identifier: NCT05342519) is currently underway to evaluate the efficacy of topical rapamycin application at 0.1% in vitiligo.

In addition, nanoparticles containing rapamycin and the autoantigen HEL46-61 (NPHEL46-61/Rapa) were synthesized, the administration of which halted disease progression in mice [94].

#### 3.1.4. Neural Hypothesis

The neural hypothesis points to neurochemical mediators secreted from cutaneous nerve endings as responsible for the melanocytes’ cytotoxic injury and death. Several clinical observations suggest this communication between the nervous system and the skin [95].

Firstly, the almost dermatomal distribution of vitiligo patches in the segmental type of vitiligo and the symmetrical distribution in nonsegmental vitiligo. Actually, in the segmental type, vitiligo is confined to one segment of the body, but it is rarely strictly dermatomal but strays to adjacent dermatomes [96,97].

Furthermore, vitiligo is described to develop in patients affected by transverse myelitis, diabetic neuropathy, and in demarcated areas after nerve damage [98,99].

Finally, severe emotional stress may trigger the onset or the exacerbation of vitiligo. The mechanism is still unclear; however, it has been postulated the increases of neuroendocrine hormones and neuropeptides, the augmentation of oxidative stress and the modification of the immune tolerance system as potential processes leading to depigmentation [100].

Neural theory is corroborated by several findings implicating the role of the sympathetic nervous system, of neuropeptides and of morphologic alterations of dermal nerves.

Melanin production seems to be affected by the dysfunction of the autonomic nerve system consisting in an increased adrenergic tone and in a decreased parasympathetic tone that leads to a three times higher cutaneous blood flow on segmental vitiligo lesions compared to normal skin [101]. Additionally, vitiligo patients display an increased level of norepinephrine in plasma and an increased concentration of urinary catecholamine catabolites, which are associated with disease activity. Catecholamines cause vasoconstriction, hypoxia, and overproduction of ROS, leading to melanocytes’ demise [102].

Neuropeptide Y (NPY), a neuropeptide linked to stress, is increased in vitiligo lesional and perilesional skin [97,102].

Vitiligo is also associated with significantly increased levels of nerve growth factor (NGF), which is also upregulated by stress [103].

Finally, the examination under the electron-microscope of dermal nerves of vitiligo skin showed ultrastructural changes such as increased thickness of the basement membrane of Schwann cells and minor axon degeneration [104].

### 3.2. Treatments of Vitiligo: Past, Present and Future

Traditional vitiligo treatments depend on the type and extent of the disease and the time of its onset.

In stable, non-segmental vitiligo involving less than 10% of the body surface, it is indicated to proceed with topical high-potency corticosteroid and topical calcineurin inhibitors. In cases of diffuse disease, narrow band (NB)-UVB phototherapy is recommended [71,95].

In cases of stable segmental vitiligo, it is possible to proceed either with targeted topical treatments (high-potency corticosteroids and calcineurin inhibitors) or phototherapy and surgical therapy with autologous transplantation of healthy melanocytes into the depigmented areas [71,95,105].

In rapidly progressive cases, systemic treatment with glucocorticoids more or less combined with NB-UVB is indicated [71,95,105].

As was mentioned earlier, new pathogenetic knowledge has allowed the development of several therapies for vitiligo. The newest ones will be reviewed here.

#### 3.2.1. Afamelanotide

Afamelanotide is a synthetic analog of the key melanogenesis molecule α-MSH that has a longer half-life and higher affinity for the target melanocortin 1 receptor (MC1R). By acting on this receptor, afamelanotide not only stimulates melanogenesis and promotes the transfer of eumelanin into melanosomes but because inflammatory cells, such as neutrophils or lymphocytes, can express the melanocortin 1 receptor (MC1R), it also acts on the altered inflammatory microenvironment of vitiligo lesions [105]. The drug is administered as a subcutaneous, biodegradable, controlled-release implant [106].

Few studies have been conducted regarding the efficacy and safety of afamelanotide in vitiligo. The most significant of these is certainly the randomized comparative multicenter trial carried out by Lim et al. The authors recruited 18 years or older patients with Fitzpatrick skin phototypes III to VI and a confirmed diagnosis of nonsegmental vitiligo involving 15% to 50% of total BSA (Body Surface Area) [107]. 28 patients were randomized to combination therapy consisting of afamelanotide plus narrowband UVB phototherapy (NB-UVB) and 27 into monotherapy with NB-UVB phototherapy. Both groups underwent 1 month of NB-UVB phototherapy, then the first group received afamelanotide 16 mg per month for 4 months while it continued NB-UVB phototherapy; the second group only continued NB-UVB phototherapy. Combination therapy of afamelanotide plus phototherapy was found to be superior (48.64% re-pigmentation) to phototherapy alone (33.26% re-pigmentation) (*p* < 0.05). In particular, greater and faster re-pigmentation was achieved in the combination therapy group for the face and upper extremities, which are the most exposed and visible areas of skin.; faster re-pigmentation was found in subjects of phototype IV-VI undergoing combination therapy than in those undergoing monotherapies, while there were no differences between the two groups in the case of phototype III. Treatment resulted in few side effects, notably erythema, nausea, and generalized skin hyperpigmentation, and generally well tolerated, although two patients dropped out of the trial because they considered the hyperpigmentation socially disabling [108].

A similar study, performed a few years earlier and on a smaller sample of patients, was conducted by Grimes et al. The authors presented preliminary results of 4 patients with generalized vitiligo who underwent one month of NB-UVB phototherapy. From the second month they received a series of 4 monthly subcutaneous implants of 16 mg afamelanotide. All patients showed areas of follicular and confluent re-pigmentation within 2 days to 4 weeks after the first administration of afamelanotide, and the improvement progressed throughout treatment [109].

Finally, in a tertiary dermatology center in Singapore, an additional randomized controlled trial was conducted to test afamelanotide in combination with NB-UVB phototherapy in Asian patients (afamelanotide plus NB-UVB implants vs. placebo plus NB-UVB implants); the study design was subsequently changed to open-label and the results were reported by Toh JJh et al. In the 18 patients enrolled and receiving biweekly NB-UVB phototherapy and monthly administrations of afamelanotide for 7 months, combination therapy was superior to placebo, with a statistically significant reduction in the median Vitiligo Area Score Index scores for total, head and neck, hands, upper extremities, trunk, and lower extremities at day 140 and later [110].

As highlighted above, the effect of afamelanotide would be higher in dark-skinned patients, which can be explained by the potent MC1R response in these patients. Clinuvel completed another similar trial in Europe in December 2012 in patients with lower phototypes, but the results have not been published or at least do not appear on clinicaltrials.gov [111]. However, another experimental clinical trial, also conducted by Clinuvel, is ongoing, whose purpose is to evaluate the efficacy and safety of afamelanotide in facial vitiligo, which is expected to close in August 2023 (ClinicalTrials.gov Identifier: NCT05210582).

In conclusion, more data is needed to adequately evaluate the efficacy and safety of afamelanotide. The molecule, at present, is certainly promising as a combination therapy with phototherapy. Only phototherapy, in fact, is able to induce melanoblast differentiation. Afamelanotide can only increase the rate and extent of re-pigmentation in subjects who respond to phototherapy, but it cannot induce melanoblast differentiation and increase the response rate. In addition, there needs to be more information on the ideal dosage and frequency of administration, the long-term response to the molecule, and its effect on phototypes I and II. Further studies are needed to confirm those examined by our review and to highlight other potentials and limitations of this molecule [112,113].

#### 3.2.2. Prostaglandins and Analogues

Prostaglandins (PGs) are polyunsaturated essential fatty acids released from cell membrane phospholipids and involved in melanin synthesis [114].

Prostaglandin F2 (PGF2α) and prostaglandin E2 (PGE2) and are the major PGs: they are synthesized in the skin and act on keratinocytes, Langerhans cells and melanocytes, stimulating melanocyte proliferation and affecting their responsiveness to neural stimulation. In addition, they stimulate the activity and the expression of tyrosinase, the rate-limiting enzyme for melanin synthesis [86,114].

On the contrary, in patients with vitiligo, PGE2 production is reduced because of oxidative stress, which causes the destruction of melanocytes and the depletion of glutathione [115].

According to this preclinical rationale, patients with focal and stable vitiligo (patches < 5% body surface area) may benefit from topic treatment with PGE2. First, in a serial series of 56 patients with stable vitiligo treated with PGE2 0.25 mg/g gel twice daily for 6 months, the re-pigmentation was observed in 40 of them (71%), with 22 (39%) and 8 (14%) patients experiencing an excellent and complete re-pigmentation, respectively. Lip burning was the only significant side effect observed in only 10% of patients [114].

The idea of using topical bimatoprost, an analogue of prostaglandin F2-alpha (PGF2α), approved for the treatment of glaucoma and inadequate or insufficient eyelashes, came from the observation of the side effect of hyperpigmentation of the periocular skin, caused by increased melanogenesis, in treated patients. Thus, Kapur et al. found a marked increase in the number of melanin granules in the skin biopsies of 2 patients [116].

Nrang et al. tested the effect of bimatoprost 0.03% ophthalmic solution in vitiligo in 10 patients; the drug was applied twice daily for four months. Of 10 patients, 3 had 100% re-pigmentation, 3 had 75 to 99%, and 1 had 50 to 75% re-pigmentation. The face was the best-responding body area [117].

Furthermore, a randomized, single-blind, individualized control study evaluated the efficacy and safety of bimatoprost 0.01% solution compared with tacrolimus 0.1% ointment, applied twice daily for 12 weeks, in 16 patients with 2 or more facial vitiligo patches. Only 10 patients completed the study. At week 12, a statistically significant reduction in vitiligo surface area in both groups compared to baseline was noticed (*p* < 0.05), without any statistically significant difference among the two groups [118].

Another randomized, double-blind, controlled proof-of-concept trial evaluated the efficacy and the safety of bimatoprost 0.03% monotherapy and in combination with mometasone, compared with mometasone plus placebo, in patients with non-segmental vitiligo or non-facial areas. Thirty-two patients were enrolled and at 20 weeks none achieved the primary endpoint of 50–75% re-pigmentation. However, the post-hoc analysis using 25–50% re-pigmentation as the response, showed that the patients treated with bimatoprost alone or combined with mometasone achieved greater re-pigmentation in the neck and trunk than patients on mometasone monotherapy [119].

Similarly, latanoprost, a PGF2α analog used to treat glaucoma, which is able to induce skin pigmentation in guinea pigs, was evaluated as topical formulation in patients with vitiligo [120,121,122].

A first study compared in the same patients topical latanoprost with placebo (group A), topical latanoprost with narrow-band ultraviolet B (NB-UVB) (group B), and their combination with NB-UVB. 22 patients with stable bilateral symmetrical vitiligo lesions for 3 months were enrolled and, after 3 months, the grade and degree of re-pigmentation were evaluated. Latanoprost was found to be better than placebo and comparable with the NB-UVB alone in inducing skin re-pigmentation. In addition, the combination of Latanoprost with NB-UVB was more effective than the use of NB-UVB alone (*p* < 0.05) [120].

Furthermore, a randomized, double-blind comparative study evaluated the therapeutic efficacy of microneedling in combination with NB-UVB phototherapy versus their combination with latanoprost 0.005% solution in vitiligo. Fifty patients with bilateral localized, stable, nonsegmental vitiligo were enrolled. Two bilateral, nearly symmetrical lesions were identified and treated with micro-needling (12 sessions at 2-week intervals) followed by latanoprost 0.005% solution on one side and placebo (saline solution) on the other side. All patients then received NB-UVB for 6 months. Both treatment regimens showed clinical improvement from baseline with a significant increase in re-pigmentation (*p* < 0.001), with a more significant degree of clinical improvement in patients receiving latanoprost (*p* < 0.002). The degree of clinical improvement was significantly higher in vitiligo lesions located on the face, neck and trunk, compared with those located on the extremities or acral sites. Side effects reported from assisted administration of latanoprost by microneedling were few and minor and no systemic side effects occurred in any patient [121].

Moreover, a pilot study enrolling 24 patients with vitiligo, evaluated the efficacy of latanoprost, compared with tacrolimus, in combination with NB-UVB and microneedling in re-pigmentation of non-segmental vitiligo lesions was examined. The data show that the activity of latanoprost in inducing re-pigmentation was comparable to tacrolimus [122].

As far as ongoing trials assessing the activity of topical prostaglandins, a clinical trial (ClinicalTrials.gov Identifier: NCT05513924) is randomizing 40 patients with localized stable vitiligo to receive topical 5-fluorouracil or topical latanoprost after skin microneedling. The primary endpoint is the clinical re-pigmentation changes of vitiligo lesions according to Physician’s Global Assessment.

In conclusion, topical prostaglandins represent an additional topical treatment option for patients with vitiligo. Current evidence suggests that their activity is higher in combination with multimodal approaches, such as skin microneedling and/or NB-UVB phototherapy.

#### 3.2.3. Janus Kinase Inhibitors

The Janus Kinase family includes JAK1, JAK2, and TYK2 implicated in the JAK/STAT signal transduction mechanism. It plays a major role in mediating numerous extracellular signals that regulate proliferative activity, differentiation, and cell migration. Inhibitors of JAK-STAT stimulate Sonic Hedgehog and Wnt signaling implicated in epidermal pigmentation and in particular the migration, proliferation and differentiation of melanocytes [123,124,125,126,127].

In addition, INF-γ has been shown to induce the production of CXCL10, which promotes the migration of autoreactive T cells into the skin [59]. Therefore, since JAK inhibitors have been shown to block IFN-γ signaling, they should prevent the accumulation of CD8+ T cells and depigmentation of lesions [58,124,125].

Tofacitinib and ruxolitinib are two JAK inhibitors used for treating rheumatoid arthritis and myelofibrosis, respectively, and there are case reports demonstrating their efficacy in vitiligo [126,127]. In addition, Phan et al. conducted a pooled analysis of nine studies. They showed that out of 45 patients treated with tofacitinib and ruxolitinib, 26 patients (58%) had a good response (>50% re-pigmentation), 10 (22%) had a partial response (<50%/some re-pigmentation), and 9 (20%) did not respond to treatment [128]. The best response was obtained for facial vitiligo. Moreover, the efficacy of tofacitinib in topical formulation has also been demonstrated in mouse models. Notably, treatment of mice with vitiligo was carried out in a short time, thus reducing the entire therapeutic dose. This may indicate that the use of these dermal and transdermal delivery systems may be helpful in reducing drug side effects and treatment cost [129].

In a preliminary open-label study, topical ruxolitinib exhibited efficacy for re-pigmentation in vitiligo of the face [130]. In a subsequent randomized trial, the efficacy of ruxolitinib cream (1.5% twice daily, 1.5% once daily, 0.5% once daily or 0.15% once daily) vs. placebo was compared in 157 adult patients affected by vitiligo with a BSA of a minimum of 3% and facial involvement of at least 0.5%. [131]. After 24 weeks, both patients receiving ruxolitinib 1.5% twice daily and once daily achieved the primary endpoint of 50% improvement of facial vitiligo area score index compared with vehicle (45, 50, and 3%, respectively). Treatment was generally well tolerated; itching at the application site and acne were the most frequent treatment-related adverse events with a frequency of 3–19 percent and 3–18 percent, respectively. Ruxolitinib is currently the only JAK inhibitor approved by the Food and Drud Administration (FDA) for the treatment of non-segmental vitiligo in patients over 12 years of age.

Baricitinib is a selective JAK1/2 inhibitor approved for the treatment of rheumatoid arthritis and atopic dermatitis [123]. It inhibits the signal transduction of several proinflammatory cytokines. To date, there is only one clinical case describing re-pigmentation in patients with vitiligo using baricitinib 4 mg daily. Currently, there is an ongoing phase 2 study in which patients received a combination therapy of baricitinib 4 mg/d and phototherapy [132].

Ifidancitinib is another dual JAK1/3 inhibitor used for the treatment of alopecia areata. Currently in phase II clinical trial for the treatment of vitiligo [124].

Ritlecitinib, an irreversible JAK3 and tyrosine kinase inhibitor, is currently used for the treatment of rheumatoid arthritis. It, along with Brepocitinib, a TYK2/JAK1 inhibitor, are currently being evaluated for their efficacy and safety profile in active vitiligo in combination with phototherapy [124].

Cerdulatinib, a dual SYK/JAK kinase inhibitor, has been evaluated for its safety and tolerability in the treatment of vitiligo in topical training (0.37% cerudulatinib gel BID) [124].

Delgocitinib is a JAK inhibitor that has been shown to be effective in the treatment of vitiligo in two cases reported in the literature. Specifically, in the two reported cases, superior efficacy was shown on disease located in the cervical region compared to the elbows. This may be due to the different thickness of the skin in the two districts, duration of the disease, and different sun exposure [133].

## 4. Conclusions

There is still a long way to a comprehensive understanding of the pathophysiological mechanisms that cause vitiligo. It is certain that oxidative stress plays a crucial role in melanocyte loss. Afamelanotide, by mimicking α-melanocyte-stimulating hormone (MSH), which in turn can increase Nrf2, a gene involved in protecting the cell from oxidative stress, is a promising treatment. In our opinion, the possibility of drug administration via a subcutaneous implant is an advantage for patients who do not have to take anything oral or use creams. In addition, the combination with NB-UVB could contribute to greater and faster re-pigmentation. However further studies are needed to evaluate the efficacy and safety of the molecule adequately.

Topical prostaglandins also represent an additional topical treatment option for patients with vitiligo. Indeed, oxidative stress reduces the level of prostaglandins whose activity promotes melanogenesis. Current evidence suggests their efficacy, and in our opinion, they could be very useful, for example, in treating segmental vitiligo in sensitive areas where topical steroids are contraindicated. However, it appears that their activity is greater in combination with multimodal approaches, such as skin microneedling and/or NB-UVB phototherapy.

Finally, it would appear that drugs that inhibit the JAK/STAT pathway are also effective in the disease. Indeed, it seems that JAK-STAT inhibitors stimulate Sonic Hedgehog and Wnt signaling, which is implicated in epidermal pigmentation and particularly in melanocyte migration, proliferation and differentiation. Ruxolitinib is the first JAK inhibitor approved by the FDA to address re-pigmentation in vitiligo patients: it is applied twice daily to affected areas of up to 10% of the body surface. The administration of oral JAK inhibitors would certainly be very convenient for patients and especially in cases of widespread non-segmental vitiligo. However, the cases are currently anecdotal.

Vitiligo has a complex pathogenetic mechanism in which multiple pathways are involved so combination therapies appear to be more effective than those targeting a single pathway. The more we know about this disease, the better we will be able to find therapeutic strategies.

## Figures and Tables

**Figure 1 ijms-24-04910-f001:**
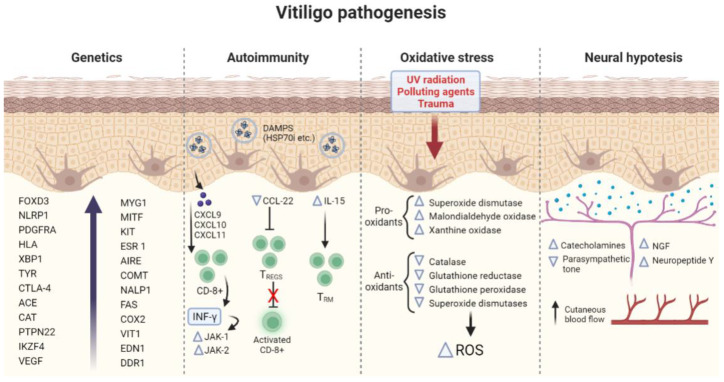
Summary of the main pathogenetic mechanisms involved in vitiligo. FOXD3 (Forkhead Box D3), NLRP1 (NLR family pyrin domain containing 1), PDGFRA (Platelet Derived Growth Factor Receptor Alpha), HLA (human leukocyte antigen), XBP1 (X-box binding protein 1) TYR (Tyrosinase), CTLA-4 (Cytotoxic T-Lymphocyte Antigen 4), ACE (Antigen converting enzyme), CAT (Catalase), PTPN22 (Protein Tyrosine Phosphatase Non-Receptor Type 22), IKZF4 (IKAROS Family Zinc Finger 4, VEGF (Vascular endothelial growth factor), MYG1 (MYG1 Exonuclease), MITF (Melanocyte Inducing Transcription Factor), KIT (KIT Proto-Oncogene, Receptor Tyrosine Kinase), ESR 1 (Estrogen Receptor 1), AIRE (Autoimmune Regulator), COMT (Catechol-O-methyltransferase), NALP1 (Nucleotide-binding oligomerization domain, Leucine-rich Repeat and Pyrin domain containing), FAS (Fas Cell Surface Death Receptor), EDN1 (Endothelin 1), COX2 (Cyclooxygenase 2), VIT1 (Vacuolar iron transporter 1), CXCL9-10-11 (C-X-C Motif Chemokine Ligand 9-10-11), DAMPs (Damage-associated molecular patterns), HSP70 (Heat Shock Protein 70 kilodaltons), CCL22 (C-C Motif Chemokine Ligand 22), IL-15 (Interleukin-15), CD8 (cluster of differentiation 8, IFN-γ (Interferon gamma), JAK1–2 (Janus Kinases 1 and 2), TREGS (regulatory T cells), TRM (Resident memory T cells), ROS (Reactive oxygen species), and NGF (Neural growth factor).

## Data Availability

Not applicable.

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
