# Peer review of "Vitiligo, from Pathogenesis to Therapeutic Advances: State of the Art"

_ijms, 2023, doi:10.3390/ijms24054910_

Round 1

Reviewer 1 Report

The author provides a narrative review of the pathogenetic features of vitiligo and the latest treatments available for the management of vitiligo. However, the structure of this article needs to be improved.

Section 2 material and method confused the reviewer about the type of this article. Is it a review or a retrospective study? For the review paper, the reviewer thinks section 2 is not necessary. On the other hand, the author must reorganize the article into an appropriate structure for a retrospective study report. 

In section 3 about the pathogenesis of vitiligo, the author may add a figure to summarize the mechanism and a table to show the relative genes.

In section 3.2 about the treatment, the author should summarize the current treatment strategy first and then go through to the latest therapies. The author should give the readers about the benefits or disadvantages of the newest treatment method. In addition, genetic issues are involved in the pathogenesis of vitiligo; the author should also discuss the polymorphisms for each treatment compound.

In conclusion, what are the author's opinions or suggestions for vitiligo, from pathogenesis to therapeutic advances, after reviewing almost 120 articles? 

The similarity rate is about 37% after excluding material and references, the author needs to rewrite it. Also, over 50% of references are before 2015. The author should provide the readers with the latest information, not the older ones.

Author Response

Dear Reviewer 1, here is the point-by-point response to your comments. Thank you for the review. Changes are highlighted in yellow. 

Answer: The author provides a narrative review of the pathogenetic features of vitiligo and the latest treatments available for the management of vitiligo. However, the structure of this article needs to be improved.

Question: Section 2 material and method confused the reviewer about the type of this article. Is it a review or a retrospective study? For the review paper, the reviewer thinks section 2 is not necessary. On the other hand, the author must reorganize the article into an appropriate structure for a retrospective study report.

Answer: I confirm that the paper is a narrative review. However, referring to the literature (Arksey et al. 2005), we believe that even in the case of narrative reviews, a research method should be followed that gives the review more dignity. Hence, the methods by which the research of sources to write the review was conducted. The article is therefore not a retrospective study.

Question: In section 3 about the pathogenesis of vitiligo, the author may add a figure to summarize the mechanism and a table to show the relative genes.

Answer: A figure containing all the pathogenetic mechanisms and genes involved disease has been included. The relevant genes have also been added to the figure.

 Question: In section 3.2 about the treatment, the author should summarize the current treatment strategy first and then go through to the latest therapies. The author should give the readers about the benefits or disadvantages of the newest treatment method. In addition, genetic issues are involved in the pathogenesis of vitiligo; the author should also discuss the polymorphisms for each treatment compound.

Answer: A section on the preclinical development of new therapies based on cytokine and genetic alterations has been added. As it is written in the introduction the aim of this review is not to examine all available treatments for vitiligo, which are nevertheless mentioned, but only to examine the newest treatments in the light of new pathogenic findings.

Question: The similarity rate is about 37% after excluding material and references, the author needs to rewrite it. Also, over 50% of references are before 2015. The author should provide the readers with the latest information, not the older ones.

Answer: It is true that about half of the references are pre-2015, but many of them represent milestones in vitiligo research and are the basis for any subsequent consideration. In addition, there are more than 50 references written beyond 2015 of which many are after 2020. Due to the risk of plagiarism, however, the manuscript was revised. However, new, updated references have been added.

Reviewer 2 Report

The authors presented a very complete and thorough review of the pathogenesis and treatments of vitiligo. The summarization of the published literature is organized and easily digestible for the reader.

In the Genetics section (lines 113-116) the authors make reference to the role of IL1-beta in the polarization of Th17 cells, I suggest the author expand the references for this to make it clearer that vitiligo (and other immune diseases) are greatly affected by Th17 cells.

In the Autoimmunity section, the authors correctly illustrate the role of the immune system in the development/progression of vitiligo. I suggest the authors add a few lines commenting on the extra evidence of the autoimmune origin vitiligo as shown in patients treated with immune checkpoint inhibitors.

The new treatments section of the review does not require major corrections; however, I ask the authors to review and if necessary, correct the reference in line 379, as it seems to be incorrectly placed. Also, the authors should double check the lines 406-408 to improve the writing of that paragraph.

Although the authors made it very clear that most of the reviewed literature refers to ongoing clinical trials, I would like to suggest the addition of a small comment on novel treatments (maybe outside of clinical trials), like immunotherapy targeting PD/PD-L1 in combination with UVB as possible treatments for vitiligo.

Finally, the authors mention both in the Janus Kinase inhibitors section as well as in the conclusions the role of JAK-STAT inhibitors in promoting proliferation and differentiation of melanocytes. Can you provide more references to further support this idea, so far the references direct to other reviews, making it harder for readers to refer directly to the papers in question.

Unfortunately, when checking the work for similarities with previous works, the similarity rate falls within an unacceptable range which requires the authors to do major rewriting of the manuscript with the corresponding adequations to the references.

Author Response

Dear Reviewer 2, here is the point-by-point response to your comments. Thank you for the review. Changes are highlighted in yellow. 

The authors presented a very complete and thorough review of the pathogenesis and treatments of vitiligo. The summarization of the published literature is organized and easily digestible for the reader.

Question: In the Genetics section (lines 113-116) the authors make reference to the role of IL1-beta in the polarization of Th17 cells, I suggest the author expand the references for this to make it clearer that vitiligo (and other immune diseases) are greatly affected by Th17 cells.

Answer: Thanks for the suggestion, a section on the involvement of IL-17 and Th17 in the pathogenesis of the disease has been added with bibliographic references.

Question: In the Autoimmunity section, the authors correctly illustrate the role of the immune system in the development/progression of vitiligo. I suggest the authors add a few lines commenting on the extra evidence of the autoimmune origin vitiligo as shown in patients treated with immune checkpoint inhibitors.

Answer: Thanks for the suggestion, the discussion of checkpoint inhibitors has been added with the relative reference.

Question: The new treatments section of the review does not require major corrections; however, I ask the authors to review and if necessary, correct the reference in line 379, as it seems to be incorrectly placed. Also, the authors should double check the lines 406-408 to improve the writing of that paragraph.

Answer: Thanks for the suggestions. The reference has been changed and the sentence on latanoprost in combination with NB-UVB has been improved.

Question: Although the authors made it very clear that most of the reviewed literature refers to ongoing clinical trials, I would like to suggest the addition of a small comment on novel treatments (maybe outside of clinical trials), like immunotherapy targeting PD/PD-L1 in combination with UVB as possible treatments for vitiligo.

Answer: Thanks for the comment, a paragraph has been added in the autoimmunity section, following the discussion on checkpoint inhibitors.

Question: Finally, the authors mention both in the Janus Kinase inhibitors section as well as in the conclusions the role of JAK-STAT inhibitors in promoting proliferation and differentiation of melanocytes. Can you provide more references to further support this idea, so far the references direct to other reviews, making it harder for readers to refer directly to the papers in question.

Answer: The reference has been added.

Round 2

Reviewer 1 Report

The author has made sufficient improvement after 1st round of review; however, the reviewer suggested that the author should summarize the current treatment strategy first and then go through to the latest therapies to provide a comprehensive view of the vitiligo treatment for the readers. After reviewing almost 120 articles, the author should also provide opinions or suggestions for vitiligo treatment in the conclusion section. 

Author Response

Question: The author has made sufficient improvement after 1st round of review; however, the reviewer suggested that the author should summarize the current treatment strategy first and then go through to the latest therapies to provide a comprehensive view of the vitiligo treatment for the readers. After reviewing almost 120 articles, the author should also provide opinions or suggestions for vitiligo treatment in the conclusion section.

Answer: Thank you for your comments. I have included a summary of current treatments in the treatment section. Given the request to add traditional treatments, the section was re-titled “Treatments of Vitiligo: past, present and future”. In the conclusion section, I have included personal remarks.

Reviewer 2 Report

The authors addressed most of the issues presented during the 1st round of reviews.

Unfortunately, the usage of automated plagiarism check tools still shows a similarity of 36% with previously published works, MDPI requires the following: "Reuse of the text that is copied from another source must be between quotes and the original source must be cited". I suggest to the authors, to use these plagiarism check tools to assess and correct this, and other issues related with "plagiarism".

In the section of Jak/STAT inhibitors, I believe the authors correctly mention the role of these inhibitors in melanocyte differentiation, etc., within the context of SONIC/Wnt signaling. However, as mentioned in the review some of these inhibitors are used for the regulation of the immune system. Because of this, I ask the author to add in this section some information on how these inhibitors regulate the immune system (e.g. CD8+, IFNs) in the context of vitiligo as an additional mechanism for controlling the disease. 

In general, the text corrections after the 1st review are deemed acceptable, but the paper cannot be published in its current for until the authors provide a major revision where the quotes and rest of the text are within an acceptable plagiarism rate. 

Author Response

The authors addressed most of the issues presented during the 1st round of reviews.

Question: Unfortunately, the usage of automated plagiarism check tools still shows a similarity of 36% with previously published works, MDPI requires the following: "Reuse of the text that is copied from another source must be between quotes and the original source must be cited". I suggest to the authors, to use these plagiarism check tools to assess and correct this, and other issues related with "plagiarism".

Answer: Thank you for your comment. The text was re-modified using the similarity report provided by MDPI via the authors' own "Grammarly" checker.

Question: In the section of Jak/STAT inhibitors, I believe the authors correctly mention the role of these inhibitors in melanocyte differentiation, etc., within the context of SONIC/Wnt signaling. However, as mentioned in the review some of these inhibitors are used for the regulation of the immune system. Because of this, I ask the author to add in this section some information on how these inhibitors regulate the immune system (e.g. CD8+, IFNs) in the context of vitiligo as an additional mechanism for controlling the disease.

Answer: Thank you for your comments, the section has been added.

Round 3

Reviewer 1 Report

The author has made sufficient improvement after two rounds of review. However, the plagiarism rate is about 37% after excluding material and references. The author needs to rewrite the manuscript or seek language editing to reduce the similarity rate.

Author Response

Question: The author has made sufficient improvement after two rounds of review. However, the plagiarism rate is about 37% after excluding material and references. The author needs to rewrite the manuscript or seek language editing to reduce the similarity rate.

Answer: Dear reviewer 1, thank you for the suggestion. I have further reduced the similarities rate.

Reviewer 2 Report

Th authors have addressed most of the issues presented to them previously. However, the anti-plagiarism software still shows high similarity to previously published works. I suggest the authors to specifically rephrase lines 166-172, 424-446 and 494-514. 

Additionally, authors should check the redaction of lines 502-505 to make it clear (I suggest inverting the order of the newly added text).

Finally, the authors should check the AIRE line in Figure legend 1, also nb in line 555 and spelling in line 574.

Author Response

Question: Th authors have addressed most of the issues presented to them previously. However, the anti-plagiarism software still shows high similarity to previously published works. I suggest the authors to specifically rephrase lines 166-172, 424-446 and 494-514.

Additionally, authors should check the redaction of lines 502-505 to make it clear (I suggest inverting the order of the newly added text).

Finally, the authors should check the AIRE line in Figure legend 1, also nb in line 555 and spelling in line 574.Kind regards.

Answer: Dear reviewer 2, thank you for the suggestion. I modified everything you requested.